# Radiopharmaceutical Treatments for Cancer Therapy, Radionuclides Characteristics, Applications, and Challenges

**DOI:** 10.3390/molecules27165231

**Published:** 2022-08-16

**Authors:** Suliman Salih, Ajnas Alkatheeri, Wijdan Alomaim, Aisyah Elliyanti

**Affiliations:** 1Radiology and Medical Imaging Department, Fatima College of Health Sciences, Abu Dhabi 3798, United Arab Emirates; 2National Cancer Institute, University of Gezira, Wad Madani 2667, Sudan; 3Nuclear Medicine Division of Radiology Department, Faculty of Medicine, Universitas Andalas, Padang 25163, Indonesia

**Keywords:** alpha particles, auger electron, beta particles, nanotargeted therapy, radioligand therapy

## Abstract

Advances in the field of molecular biology have had an impact on biomedical applications, which provide greater hope for both imaging and therapeutics. Work has been intensified on the development of radionuclides and their application in radiopharmaceuticals (RP_S_) which will certainly influence and expand therapeutic approaches in the future treatment of patients. Alpha or beta particles and Auger electrons are used for therapy purposes, and each has advantages and disadvantages. The radionuclides labeled drug delivery system will deliver the particles to the specific targeting cell. Different radioligands can be chosen to uniquely target molecular receptors or intracellular components, making them suitable for personal patient-tailored therapy in modern cancer therapy management. Advances in nanotechnology have enabled nanoparticle drug delivery systems that can allow for specific multivalent attachment of targeted molecules of antibodies, peptides, or ligands to the surface of nanoparticles for therapy and imaging purposes. This review presents fundamental radionuclide properties with particular reference to tumor biology and receptor characteristic of radiopharmaceutical targeted therapy development.

## 1. Introduction

In the early 1900s, Henri Becquerel and Marie Curie discovered radioactivity. Therapeutic applications immediately followed this discovery [1,2]. For many years radionuclide therapy was limited to the use of Iodide-131 (^131^I) for thyroid cancer and hyperthyroidism and phosphate-32 (^32^P) for polycythemia vera [2,3,4,5,6]. Radionuclides labeled molecules such as a drug, a protein, or a peptide that operate as a delivery vehicle that accumulates and binds to specific targets such as tumors or other undesirable cell proliferation [3,7,8]. The development of radionuclide use has been growing exponentially with the introduction of more new radiopharmaceuticals (RPs) for therapy and imaging.

In recent times, RPs use in nuclear medicine has become popular in theranostics. These are used in therapeutic interventions after imaging verifies the presence of a biological target [6,9,10]. Unlike radiotherapy, RPs are administrated intravenously to be delivered to a target tumor or associated structure. RPs have advantages in treating systemic malignancy in areas such as the bone or brain, which are impossible to treat using external radiotherapy [2]. The targeted tumor cell absorbs a dose of radiation from an RP which exponentially decreases over time (Figure 1a). On the other hand, in external radiotherapy, radiation beams are directed at tumor tissue and cannot avoid healthy cells (Figure 1b).

Radiopharmaceutical therapy (RPT) is a novel modality that can be effective with minimal toxicity [6,7]. The advantages of RPT are, firstly, it can be targeted at tumors, including metastasis sites. The RPs can be used in radiotracer imaging to determine the uptake of the RP in the target tissues before administering a therapeutic dose. Secondly, a wide variety of radionuclides are now available emitting different types of radiation at different energies. For instance, high linear energy transfer (LET) radionuclides are used effectively to kill resistant hypoxic cells. Thirdly, this therapy allows for a relatively lower whole-body absorbed dose [7,10,11,12,13].

RPT can be used as adjuvant therapy with or after other treatment options such as chemotherapy and surgery [2]. It is being used to control symptoms and shrink and stabilize tumors in systemic metastatic cancer, where conventional therapy or chemotherapy is impossible. RPT can be a good choice, especially for patients who no longer respond to other treatments [2,3,7,10]. This review describes some fundamental radionuclide properties with particular reference to tumor biology and the receptor characteristics of radiopharmaceutical targeted therapy development.

## 2. Radionuclide Emission Properties

The physical characteristics of a radionuclide should be considered when selecting it for therapy purposes. These include physical half-life, radiation energy, type of emissions, daughter product(s), production method, and radionuclide purity [2,9]. Ideally, the physical half-life of the radionuclide should be between 6 h and seven days [14]. The RPs with a long half-life will expose the target tumor and surrounding environment to radiation for longer. However, RPs with a very short physical half-life have limitations due to the delivery time. There must be sufficient retention time for the emission to be delivered to the tumor target [15].

Furthermore, in vivo stability, toxicity, and the biological half-life within the patient’s body must be considered [7,16], along with the type and size of the tumor, method of administration, and uptake mechanism [1,2,6,15]. The tumor uptake mechanism is specific to the target cell. It depends on processes such as antigen–antibody reactions, physical particle trapping, receptor binding sites, removal of damaged cells from circulation, and transportation of a chemical species across a cell membrane and metabolic cycle [2,17]. The condition will influence the ratio of the concentration of radionuclides in the tumor to that in normal tissues. This ratio should be optimized [2]. The other factors that must be considered are radionuclide particle size, toxicity, specific gravity for optimal flow and distribution, and clearance rate [2,6,18,19,20,21,22].

Radionuclides used in RPT are primarily beta (β)-particle (0.2 keV/μm) or alpha (α)-particle (50–230 keV/μm) emitters [2,9,11,15,23], and Auger electrons (AE) (4–26 keV/μm) [2,9,11,15,23,24]. Various radionuclides and their characteristics are summarized in Table 1. Each of these radiation types results in ionization along the travel length, and they are fully deposited in the cell [16]. The radiation destroys the cell directly and indirectly [6,25]. The distance traveled by particles and the energy deposited in cells must be considered to ensure optimal targeted cell destruction and minimize ionization interaction with healthy cells [2,6,7,15].

### 2.1. Beta Particles

Beta particles have been used in cancer therapy over the last 40 years [6]. They are the product of the β decay process, wherein an unstable nucleus is converted to a proton, and a β particle, a high-energy electron [7,26]. β particles are the most frequently used radiation in RPT agents and are widely available [7]. β particles are negatively charged. They have a relatively long path from 0.0 to 12 mm, and some emit a gamma (γ) ray such as ^32^P, ^89^Sr, ^90^Y, and ^169^Er [3]. They emit γ ray <10%, which is acceptable for imaging to confirm the tumor uptake and biodistribution and dosimetric calculations [2,3]. They have a low linear energy transfer (LET) of approximately 0.2 keV/μm, so more β particles are required to deliver a similar absorbed dose compared to alpha particles.

The most familiar and frequently used β particle is iodine-131 (^131^I). Hertz and Roberts used radioiodine I-130 (^130^I) for hyperthyroid therapy in 1941, which rose at the birth of nuclear medicine [27,28,29]. In August 1946, ^130^I was replaced by ^131^I because it was much cheaper [27,29]. ^131^I is a β and γ emitter with a half-life of 8.05 days. The β particle has a peak energy of 0.606 MeV, with a maximum range of ~3 mm in the tissue, and it is used for therapy. The peak energy of the γ ray is 0.364 MeV and is used for imaging [27]. Since then, ^131^I has been used countless times for therapy for hyperthyroid and thyroid cancer [3,6,27,28,29,30]. In 1981, ^131^I-iobenguane (meta-iodobenzylguanidine, MIBG) was introduced as a diagnostic agent, and in 1984, it was used for treating malignant phaeochromocytoma [31]. Monoclonal antibodies are used to label with ^131^I, and, in 2003, FDA approved ^131^I-tositumomab (Bexxar) for the treatment of refractory non-Hodgkin’s lymphoma (NHL) [2,6,7]. Several studies have reported the monoclonal antibodies labeled on other beta particle emitters, including Yttrium-90 (^90^Y) and Lutetium-177 (^177^Lu), for more effective therapy purposes [2,7,31,32,33].

The high-energy β from Yttrium-90 (^90^Y) or Rhenium-188 (^188^Re) is preferable for treating higher volume solid and poorly perfused tumors and is less suited for targeting micro-metastases to avoid crossfire doses to neighbor cells [9,11,34]. ^90^Y, widely available like ^131^I, is a popular radionuclide for liver cancer and metastases [35,36]. Neuroendocrine tumors (NET_S_) have been treated with radionuclide therapy (PPRT) targeting peptide receptors with radiopharmaceuticals labeled with ^90^Y. Antibodies also labeled with ^90^Y, have been introduced for ovarian and hematological cancers [7,26,37,38,39]. Low-energy ẞ, like those seen with lutetium-177 (^177^Lu), is more efficient for small tumors [1,9]; hence, ^177^Lu is becoming a popular ẞ-particle source for treating small tumors [7,9]. ^177^Lu has a half-life of 6.73 days and is compatible with antibodies and peptides [40,41]. Furthermore, it also emits gamma-rays and can be detected externally as a theranostic agent [1,7,40]. Samarium-153 (^153^Sm) is used to treat palliative bone metastases and other primary cancers [3,42,43]. Ethylenediamine-tetra-methylene-phosphonic acid (EDTMP) chelator binds with ^153^Sm through six ligands (four phosphate groups and two amines). It has been widely used since FDA approval in various osteoblastic metastatic lesions, especially in prostate and breast cancer [44]. However, not all possible β particle sources have been widely adopted because of the complexity of the radiochemistry or the absence of commercial availability. The decision to use one β-particle source over another must consider the absorbed dose ratio between tumor to non-tumor tissue [7].

### 2.2. Alpha Particles

The application of targeted α particle therapy (TAT) gained approval in 2013 [19]. Alpha particles are high energy and have shorter path lengths, resulting in higher efficacy in some applications [2,8,15,25,26]. TAT is an attractive therapeutic option for multiple micro-metastases. It is easy to administer and can be used to treat multiple lesions simultaneously. It is also possible to combine it with other therapeutic approaches, primarily for cancer treatment [45,46].

An alpha particle is a ^4^He nucleus without its surrounding electrons (sometimes denoted as (He^2+^)) [26,45]. Alpha radiation is emitted from the nucleus of a radioactive atom undergoing decay with an energy is 4–9 MeV, and the particles travel only 1–3 cell diameters (40–100 μm) in tissue [7,15,32,45,46]. The particles have high LET (60–230 keV/μm) throughout their range, peaking to three times the initial value at the end of the path range (the Bragg peak) [16,26,32]. Most alpha particles also emit gamma-ray. However, treatment planning or post-therapeutic imaging using alpha particles is not performed yet in clinical settings due to technical limitations [45].

Furthermore, intracellular accumulation of the α particles effectively creates double-strand breaks (DSBs) in DNA, and numerous clusters of DSBs in target cells, making cellular repair systems ineffectual [7,32,47]. The cytotoxicity of α-particles is much higher than that of β-particles due to the particle deposit energy per unit path length, which is 1500 times more than beta particles [45,48]. In addition, the short travel distance of α particles reduces the damage to surrounding healthy tissue [15,49]. The particle radiation has been demonstrated to be independent of cell oxygen concentration [15,32,45,50]. The physical and biological characteristics of alpha, beta particles, and Auger electrons are summarized in Table 2, and DNA damage by that radiations are illustrated in Figure 2a,b.

Improvements in understanding molecular tumor biology, labeling techniques, technology development, and other related disciplines have paved the way for significant new clinical applications of α radiation as a novel therapeutic agent [7,15,51]. Alpha particle-labeled biological molecules such as monoclonal antibodies (mAb) allow close radiation targeting and selectively deliver high radiation to the target, with limited toxicity to normal tissues [15]. The mAbs are labeled radionuclides that bind to the extracellular domain of PSMA, demonstrating promising results in imaging and therapy of prostate cancers [9]. The monoclonal antibodies are labeled with bismuth-213 (^213^Bi) and astatine-211 (^211^At) and are used to treat leukemia and brain tumors [11,52]. The monoclonal antibody MX35 labeled ^213^Bi successfully treated ovarian cancer in animal models with no signs of toxicity [53]. ^213^Bi has a short half-life and is produced using a generator and labeling to produce TAT compounds is therefore completed on-site [26,54]. Because of its short half-life, ^213^Bi needs to be delivered directly into tumor tissue, and it can be given at a high dose over a short period, which is more effective than low dose rates given over a more extended period [26,32,55]. ^213^Bi has been used to label DOTA peptides in preclinical and clinical trials with >99% purity [15,26]. In preclinical and clinical studies, ^213^Bi and ^225^Ac have been used to label somatostatin receptors [15,26,32].

Radium-223 dichloride (Xofigo), a α particle emitter used for bone pain palliation in prostate and breast cancer patients, was approved by FDA in 2013 [7,11,26,32,45]. The emission energy of ^223^Ra can generate irreparable DNA double-strand breaks in the adjacent osteoblasts and osteoclasts, which has a detrimental effect on the adjacent cells and inhibits abnormal bone formation [7]. ^223^Ra is being studied as a radioactive label for other cytotoxic agents such as poly (ADP-ribose) polymerase inhibitors (olaparib), docetaxel (DORA trial), and new androgen axis inhibitors as enzalutamide and abiraterone citrate. The recently high number of ^223^Ra and in combination with other therapeutics, showed promising results [7].

Another alpha particle attracting increasing interest is ^225^Ac, the parent of ^213^Bi, which is relatively long-lived, with a half-life of 9.9 days [54]. ^255^Ac is produced via the neutron transmutation of ^225^Ra or decay of ^233^U [26,54,55]. ^225^Ac can be used to treat neuroendocrine tumors. It has been used to label PSMA with a radiochemical purity of >98% for prostate cancer therapy [26,54,56]. It also labeled antibodies to test for myeloid malignancy [9] and shows a potential for therapy, and post-therapy imaging, even though the images are suboptimal [26,32,55]. Results of clinical trials using TAT results indicate that this treatment strategy presents a promising alternative to targeted cancer therapy [52]. Lately, ^225^Ac-labeled PSMA-ligands have gained popularity as an alternative to ^177^Lu-PSMA [26,54,56]. However, ^225^Ac may damage the healthy cells due to daughter radionuclides such as ^221^Fr, ^217^At, and ^213^Bi [47]. Danger from radiation from daughter radionuclides needs to be carefully evaluated.

**Table 2 molecules-27-05231-t002:** Physical and biological characteristics of α, β particles, and Auger electron.

	Alpha Particle	Beta Particle	Auger Electron
Type of particles	^4^He nucleus	Energetic electron	Low energy electron; electron capture (ec) and/or internal conversion (ic)
Particle energy	4–9 MeV	50–2300 keV	25–80 keV
Particle path length	40–100 μm	0.05–12 mm	Nanomicrometers
Linear energy transfer	~80 keV/μm	~0.2 keV/μm	4–26 keV/μm
Hypoxic tumors	Effective	Less effective	Effective
Toxicity	Effective in creating double-strand breaks in DNA	High dose rates (tumor survival rates close to linear exponential). Low dose rates (single-strand breaks), repairable with shouldering the dose-response curve	Potential creation of double-strand breaks DNA, and cell membrane
Bystander effect/crossfire	Yes/low	Yes	Yes
Tumor size	Micro/small	Higher volume solid tumor	Micro

Ref: [7,8,13,24,25,26,32,47,50,55,57].

**Figure 2 molecules-27-05231-f002:**
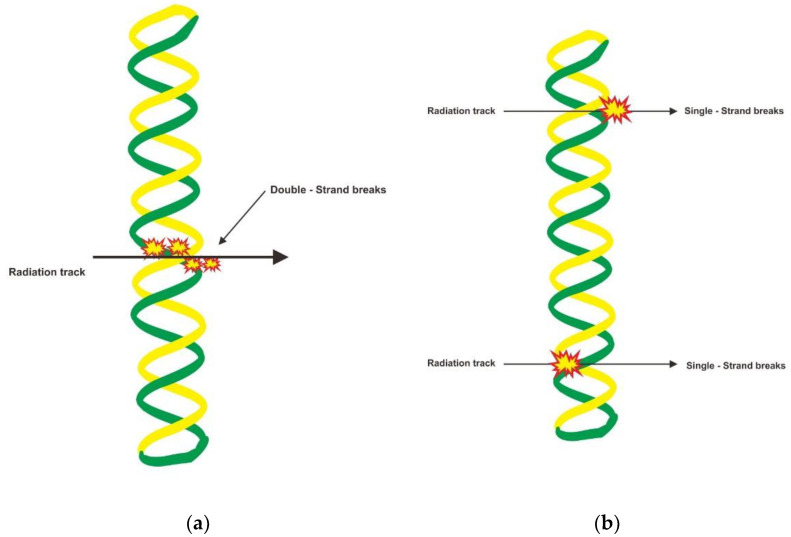
High and intermediate LET radiation (alpha particle and Auger electron, respectively), cause double-strand breaks in DNA (**a**). Single-strand breaks in DNA due to radiation by low LET (beta particle) (**b**).

### 2.3. Auger Electrons

Auger electrons (AE) have an even shorter range than alpha particles delivering radiation of only 1–1000 nm through the tissue causing potent tumor cell death if they can be conjugated with suitable ligands that effectively target micro-metastasis, particularly of DNA and cell membranes [2,11,24,32,34,52]. AEs are generated from suborbital transitions, and their range depends on their energy. They have an intermediate LET (4–26 keV/μM) [4,32,58]. Bromine-77 (^77^Br), indium-111 (^111^In), iodine-123 (^123^I), and iodine-125 (^125^I) are the most commonly used radionuclides sources [24,59,60]. Human studies using locoregional administration have shown promising results in therapy [7].

Despite the short range of AE, local cross-dose effects occur in cells adjacent to the radionuclide decay mediated by the several micrometer ranges of higher energy AEs and internal conversion (IC) that causes the death of distant non-irradiated cells through the bystander effect [16,24,25,27,32]. Lethally damaged tumor cells may release mediators that cause the death of distant non-irradiated cells [24,25]. Radiation also releases heat shock protein-70 and high mobility group box-1, which activate the dendritic cells (DCs). The activated DCs activate cytotoxic T cells that result in tumor regression at distant sites [11,61]. It has been observed that the effects of ionizing radiation can work synergistically with targeted immune treatment observed at the site(s) distant from targeted tissues/organs. This phenomenon is suggestive of the role of the abscopal effect [11,25,61,62]. Attention has been focused on delivering AE to the nucleus/DNA as the primary cellular radiation target to maximize toxic effects. However, cell membrane targeting has also been proven to be an effective strategy for killing cancer cells [24,63]. Cell membrane damage further induces γH2AX foci in the nucleus of the cells exposed to ^125^I-anti-CEA mAbs and in non-exposed cells through a bystander effect. ^125^I-labeled anti-CEA 35A7 was also found to be effective in vivo for treating small peritoneal tumors in mice [24,63]. Toxicity may also be induced indirectly by free radical-mediated pathways [24,25,57].

So, AE nuclear targeting is essential but not always required for RPT [24]. The abscopal and indirect killing effects suggest that targeting cell surface antigens overexpressed on tumor cells that are recognized by monoclonal antibodies (Mabs) or other ligands may be effective [25,64]. AE therapy has not been widely adopted yet. The fact that auger electron agents are incorporated into the DNA, and the cell membrane results in unfavorable pharmacokinetics, might be the reason for the lack of efficacy. Technological developments could overcome obstacles and increase interest in AE for therapy development [7].

## 3. Therapy Application

Radiopharmaceutical or radioligand therapy includes systemic radiation therapy, molecular radiotherapy, targeted radiation therapy, or peptide receptor radionuclide therapy (PRRT) and there are examples of where RPT is applied in optimizing and balancing the therapeutic index (TI). Various radioligands are being developed and investigated to target molecular receptors or intracellular components in personal therapy [58].

### 3.1. Antibodies

The monoclonal antibodies (mAbs) are labeled with radionuclides. The smaller fragments and new fusion proteins are directed against tumor antigens to deliver radionuclides to the targeted tumor [2,9]. The FDA has approved these agents for the clinical management of liquid malignancies (ibritumomab tiuxetan (Zevalin) labeled with ^90^Y, and tositumomab (Bexxar) labeled with ^131^I) is used for lymphoma therapy [2,6,7,55,64,65], and some the RPT in optimizing and balancing the therapeutic index (TI) [65]. The therapeutic benefit is achieved when the cells absorb continuous radiation emitted by radionuclides tagged to mAbs while minimizing toxicities in non-target tissues.

The effect of the RPT depends on the radiation’s energy and the antibody’s affinity, antigen target concentration on the cells, tissue vascularity, and antibody/antigen rate constants [64]. Novel antibody engineering techniques have enabled the development of antibodies that bind to antigens expressed in target cancer cells. An antibody that binds to a particular antigen will allow for a higher RP uptake within tumor tissue. However, antibodies are larger molecules, limiting the tumor penetration and distribution of the radiolabeled antibody within the tumor. Furthermore, antibodies have a prolonged circulation time and slow biological clearance, leading to larger radiation-absorbed doses to healthy organs and blood. Pre-infusion of a certain mass of non-radiolabeled antibody (cold antibody) may be used before the infusion of radiolabeled (hot antibody) to saturate antigenic sites in normal cells to avoid unnecessary radiation to healthy cells [64], reducing the binding of the hot antibody and decreasing the radiation doses to healthy organs. However, a pre-infusion time before administration of the hot antibody must be determined and optimized for every therapy [64].

Patient selection for RPT should be based on the predetermined expression of specific tumor antigens or diagnostic results [65]. Several antigens or receptors are expressed on the surface of the membrane of tumor cells, such as human epidermal growth factor receptor 2+ (HER2+), epidermal growth factor receptor (EGFR), CD20, prostate-specific membrane antigen (PSMA), vascular endothelial growth factor (VEGF), mucin 1 (MUC1) and tumor necrosis factor (TNF). Any of these can be labeled with various radionuclides [9]. Beta particle emitters have often been labeled with antibodies because they emit β and γ rays and have a longer half-life of 8 days. Lately, alpha particles have rapidly gained interest and have been used to label antibodies to deliver radiation to tumors, such as ^227^Th-anti CD22 and ^225^Ac-PSMA-617 [55]. However, α particles cannot be imaged unless they emit γ rays as ^223^Ra and ^227^Th do [64]. Unfortunately, these radionuclides only emit γ rays in low concentrations, which is not optimal for assessment. This imaging limitation may lead to noncompliance, and other radionuclides imaging may be required to establish lesion targeting and dosimetry [45,64].

Radionuclide-labeled mAbs demonstrate more efficacy in inducing cancer remissions than unlabeled molecules and are also more effective than chemotherapy [9]. They have been shown to benefit lung, pancreatic, stomach, ovarian, breast, colorectal, leukemia, and high-grade brain glioma cancers [2]. Fortunately, the application of the RPT in giant solid tumors is less successful than in small volume tumors such as malignant lymphoma due to poor perfusion, increased intratumoral hydrostatic pressure, and various radionuclides uptakes by the cells [8,64,65].

### 3.2. Prostate-Specific Membrane Antigen (PSMA)

^131^I-labeled prostate-specific membrane antigen (PSMA) ligands showed promise for prostate cancer therapy and were further developed to ”the ^177^Lu-PSMA” introduced in 2015 [60]. PSMA is a transmembrane protein that is over-expressed in prostate cancer (PC) cells, and its expression increases progressively in higher-grade cancers such as metastatic castration-resistant prostate cancer (mCRPC) PC [56,66,67,68]. Its benefits remain high even after multiple lines of therapy [56,66]. Radionuclide PSMA is a promising therapeutic approach for mCRPC patients for whom chemotherapy has been ineffective [55,56,66,69]. Early reports show that ^177^Lu-PSMA is safe and effectively reduces the tumor burden. It has low toxicity [69] and has become popular, with more than a thousand therapy cycles performed [66,69]. Severe hematological side effects are rare. Organs at risk after treatment with ^177^Lu-PSMA, including the salivary glands and the kidneys. However, the radiation dose to bone marrow, spleen, and liver is below critical limits [68].

Currently, the two most frequently used PSMA ligands are PSMA-617 and PSMA-I&T (imaging and therapy), labeled with ^177^Lu [68]. PSMA- targeting ligands using ^225^Ac maybe have an advantage compared to PSMA-targeting ligands using β particles. Clinical studies using ^225^Ac-labeled PSMA-ligands (PSMA-617 or PSMA-I&T) have demonstrated remarkable therapeutic results recently. Data on treatment with ^225^Ac-PSMA-617 indicate an excellent effect on tumor control in both early and late-stage mCRPC [70]. A novel α particle treatment with a ^227^Th-PSMA has shown potency in in vitro studies and efficacy in xenograft models of prostate cancer [8,67]. However, α particles have a more significant radiobiological effect on the organs at risk [56]. Concerns have been raised about treatment-associated, mostly permanent xerostomia, frequently leading to treatment discontinuation in many patients [56,68]. Combining α particles with β particle emitters is called “tandem therapy” and may reduce these significant adverse effects compared to using α particles alone [56,71,72].

### 3.3. Peptide Receptor Radionuclide Therapy (PRRT)

Receptor-based radionuclide therapies (PRRT) targeting the somatostatin receptor (SSTR), have since early 1990 been an important treatment modality for neuroendocrine tumors [7,26]. The efficacy of PPRT therapy might be due to the somatostatin receptor ligand that binds the specific receptor (SSTR1–5) [30,73]. Peptide receptors expressed in various tumor cells, including NETs, are significantly higher than in normal tissues or cells. NETs overexpress the SSTR2 potential for SSTR2 targeted therapies such as synthetic somatostatin analogs (SSAs) and radio-peptides or PRRT [30,73], and SSTR2 is primarily targeted by PRRT [73]. Octreotide and lanreotide are two SSAs developed and employed for clinical practice, which bind primarily to SSTR2 and SSTR5 [73]. Peptides have been labeled with several radionuclides, such as beta particles emitter ^177^Lu and ^90^Y. ^177^Lu–SSTR ligand is more effective in small-sized tumors, whereas, for larger tumor volumes, ^90^Y might be a better choice [30,73]. The first agent used was ^90^Y-labeled DOTATOC and DOTATATE. However, significant permanent kidney damage has been reported [34,74]. ^177^Lu-labeled DOTATATE or DOTATOC was the next PRRT radiopharmaceutical, causing less nephrotoxicity compared to ^90^Y [26] and a more negligible crossfire effect, particularly on small and metastatic tumors [74]. ^177^Lu-DOTATATE (Lutathera^®^) has also become the most widely used PRRT radiopharmaceutical at present [34].

Overall, α-emitters PRRT has shown good results. However, crossfire effects on small-size tumors have a significant impact. Additionally, hypoxia tumor tissue could be resistant to β-emitters treatment. α particles with high LET over a short range can minimize damage to surrounding healthy tissue. ^213^Bi and ^225^Ac have been clinically tested for brain tumors, neuroendocrine tumors, and prostate cancer therapy [26]. ^213^Bi and ^225^Ac-DOTA chelated peptides have been developed for peptide receptor radiotherapies, such as DOTA-Substance P targeting the neurokinin-1 receptor and somatostatin-analogs (e.g., DO-TATOC, DOTATATE) [74]. However, the results from these agents need to be confirmed in further studies.

### 3.4. Radioiodine Concentration via Sodium Iodide Symporter

^131^I has been used for adjuvant therapy to manage well-differentiated thyroid cancer (DTC) for more than 60 years. It is used to destroy remaining thyroid cells post-thyroidectomy, including in metastases, and is relatively inexpensive and widely available [14,75]. It increases the 10-year survival rate to 80% and decreases mortality by 12% [75]. One-third of advanced DTC metastases show low uptake of iodine. Losing the ability to accumulate iodine can occur during the progression of the disease due to dysfunction and loss of sodium iodide symporter (NIS) expression [75,76], indicating a status of dedifferentiation known as a radioiodine refractory disease [75,76,77].

A sodium iodide symporter (NIS) transports iodine through the cell membrane. Iodine is transported into follicular thyroid cells against the electrochemical gradient [27,75,76]. In a normal condition, the gradient between a thyroid cell and the extracellular environment is 100:1 [27,75]. The expression of NIS provides the molecular basis of radioiodine for diagnostic and therapeutic use in patients with thyroid disease [76,77,78]. It resides in the thyroid in the basolateral membrane of epithelial cells and transports two cations of sodium (Na^+^) and one anion of iodide (I-) into the cells. This process is facilitated by an enzyme Na^+^/K^+^ ATPase [27,29,75,76].

Genetic alteration causes the mitogen-activated protein kinase (MAPK) and phosphoinositide 3-kinase (PI3K) pathways associated with the silencing of solute carrier family five-member 5 (SLC5A5), which encodes NIS. The condition causes the cancer cell failure to take radioiodine [77]. A clinical trial of kinase inhibitors targeting the MAPK or PI3K pathways has shown promising effects in redifferentiation therapy. It brings hope to future therapy using either kinase inhibitors with different targets or kinase inhibitors and ^131^I in managing radioiodine refractory disease in DTC [77].

Furthermore, NIS transgene has been successfully transferred selectively into extra-thyroidal tumor cells or cells in the tumor environment using various gene delivery systems [78]. An advanced endogenous PDAC mouse model study indicated genetically engineered mesenchymal stem cells (MSC) as NIS gene delivery vehicles demonstrate high stromal targeting of NIS by selective recruitment of NIS-MSCs after systemic application resulting in an impressive ^131^I therapeutic effect [78].

### 3.5. Nanotargeted Radionuclides

In the last three decades, there has been a rapid increase in the use of new nanomaterials and radionuclides to enhance cancer diagnosis and therapies [2,79]. Many organic and inorganic materials can be used as nanoparticles [58,80,81]. Nanoparticle (NP) delivery systems have enhanced imaging and therapeutic efficacy by targeting the delivery of radio-labeled drugs to the tumor site and reducing their toxic side effects [79,81,82]. The significant advantages of nanoparticles are that they can be prepared in sizes <100 nm. This increases the localization of the drugs and radionuclides and the permeability and retention (EPR) effect of passive targeting tumor cells and facilitates uptake by active targeting tumor cells [81,82]. The surface of nanomaterials is usually coated with polymers or ligands to improve biocompatibility and the selection of specific targets [80]. A nanomaterial’s final size and structure depend on the salt concentrations, surfactant additives, reactant concentrations, reaction temperatures, and solvent conditions used during synthesis [79,80]. Two mechanisms of nanoparticle delivery system for diagnostics and therapy to tumor sites are (i) specific passive targeting cells and (ii) specific active targeting cells [81,82].

Nanotargeted radionuclides have three main components, the nanoparticle core, the targeting biomolecule (which must be able to recognize a specific biological target), and the radionuclide [80]. Nanoparticles drug delivery systems can be made from polymers (polymeric nanoparticles, micelles, or dendrimers), lipids (liposomes), viruses (viral nanoparticles), organometallic compounds (nanotubes), inorganic nanoparticles (fullerenes, carbon nanotubes, quantum dots, or magnetic nanoparticles) [47,80,81,82]. The physical and chemical properties of nanoparticles play a critical role in determining particle–cell interactions, cellular trafficking mechanisms, biodistribution, pharmacokinetics, and optical properties [80]. Each nanoparticle type shows certain advantages and disadvantages that are inherent features of a particular material, such as solubility, thermal conductivity, ability to bind biomolecules or linkers, chemical stability, and capacity to incorporate and release compounds, as well as biocompatibility, toxicity, immunogenicity, and controlled drug release rate [47,80].

The targeting biomolecule must have a high affinity for the targeted epitopes. For ligands to bind effectively, each radionuclide can be conjugated directly on the nanoparticle surface, with or without a spacer, or can be attached to the nanoparticle during chemical synthesis. The spacer groups between the nanoparticle surface, the radionuclide, or the biomolecule can be a simple hydrocarbon chain, a peptide sequence, or a PEG linker [80,81,82]. In some cases, a bifunctional chelating group (BFC), such as 1, 4, 7, 10-tetraazadodecane-DOTA, must be conjugated to the nanoparticle, and then a radioactive metal needs to be attached. This requires modification of nanoparticles before radiolabeling [80].

There are many passive and active targeted nanoparticle therapies being developed. Most development is still at the in vitro or animal study stage. The most significant development is of ^131^I labeled nanoparticles for targeted therapy of different tumor types, according to the targeting strategy of the prepared NPs in which ^131^I is incorporated. The targeting strategy of these NPs depends on either passive or active targeting. ^131^I labeled NPs (silver or polymeric) shows ^131^I accumulation in different tumor types [79]. ^131^I labeled NPs targeting integrin have been studied. This protein is essential in regulating angiogenesis processes and tumor progression. Radionuclide labeled arginine–glycine–aspartate (RGD) can specifically target tumor integrin receptors [79]. Other β-particles such as ^188^Re, Holmium-166 (^166^Ho), ^90^Y, and gold-198 (^198^Au)-NPs have also been investigated for tumor-targeted radiotherapy [79,82]. ^88^Re-liposome has been shown to have a therapeutic effect in various animal models and translational clinical research [83]. ^166^Ho nanoparticles have also been prepared and studied in radionuclide tumor therapy for skin cancer and ovarian cancer metastases [80,82]. Liposome labeled with ^90^Y has been investigated for colon and melanoma tumors in animal models [79]. Gum arabic, functionalized peptide and protein, coated ^198^Au NP have been shown to be potential prostate cancer therapeutic agents in an animal model [79].

Nanoparticles labeled with α-particle emitters have been synthesized to enhance therapeutic efficiency with minimum danger to healthy tissues. ^211^At has been studied as a prospective NP’s alpha particle emitter, but the main disadvantage of ^211^At for NTR is low in vivo stability [79]. The sodium form of A-type nano-zeolites targeting peptides has been labeled with ^223^Ra, showing a cytotoxic effect on glioma cells [79]. A preliminary study reported that ^225^Ac-Au@TADOTAGA administrated intratumoral delayed tumor growth in glioma xenografts, and it is the first reported study using ^225^Ac-labeled gold nanoparticles [84]. However, ^225^Ac use remains challenging for insufficient retention of its daughter’s products due to the α recoil effect observed upon release of an α particle [84]. So, the use of α particles still has challenges related to incorporation into useful targeting vectors, such as in vivo stability, the weakness of α emitter–biomolecule bond, organ toxicity of inappropriate leakage of radionuclides from the bioconjugate, and uncoupling (or trans-chelated) than distribution to off-target areas [47].

## 4. Challenges in Radiopharmaceutical Therapy

In the last three decades, there has been a growing interest in radioiodine, a beta and gamma emitter, as new RPs are introduced for therapy and imaging (theranostics) for specific target tumor cells. However, one must be aware of issues related to the crossfire effect and toxicity of ß particles. The high LET and short range of α particles enable effective and rapid cancer therapies but are hindered by short half-lives and rarely emit gamma radiation for imaging [9,15,55]. Combining α and β particle emitters may reduce some of these obstacles [56,64]. Some issues with RPs are related to tumor-targeting uptake, biocompatibility, side effects, nonspecific uptake and distribution, and the radiation exposure effect in healthy tissues.

The development of intelligent drug delivery agents such as peptides, small molecules, mAb, and mAb fragments, and especially nanoparticle cores offer the promise of better diagnostic and therapeutic options [55,81]. However, the heterogeneity of RP uptake by tumor cells is challenging when using radiolabeled antibodies. The larger size of whole antibodies may limit penetration into the tumor tissue and crossfire effects, which occur when radiation interacts with cells away from the site of actual binding of the antibody agent [64].

Radiopharmaceuticals provide effective cancer treatment, particularly when other standard therapeutic approaches have failed. However, even after more than four decades of clinical investigation, RPs have still not become a standard part of cancer management therapy, which is peculiar, especially in light that other “targeted therapy” have clinical trial failure rates of 97% and is more popular than RPs that [7,8,85]. Furthermore, changing the fear of the public perception of radioactivity and the perceived complexity of the treatment are challenges in developing and applying RPs for therapy and imaging.

## 5. Conclusions and Future Direction

Radiopharmaceutical therapy can be a safe and effective targeted approach to treating many types of cancer. RPT has shown high efficacy with minimal toxicity compared to other systemic cancer treatment options. Different radioligands can be chosen to uniquely target molecular receptors or intracellular components, making them suitable for personal patient-tailored therapy in modern cancer therapy management. Further research is still needed regarding specific targets, radioligand stability in vivo, toxic effects, crossfire, dosimetry, and bond stability with daughter nuclides, particularly for alpha emitters.

However, new particle drug delivery systems continue to enhance targeted therapy efficacy and safety, including the use of nanoparticles. The number of successful studies exploring new drug delivery agents’ different delivery systems of radionuclide particles will probably increase the effectiveness and range of applicants. With a growing positive track record, public understanding and perception of the safety and success of RPT may improve.

If this occurs, then RPT will be adopted as an increasingly mainstream cancer therapy approach and the investment needed to resolve issues of radionuclides supply. In the coming decades, RPT may provide an increasing variety of rapid, personalized, practical, effective, and affordable treatments that offer new hope to cancer patients.

## Figures and Tables

**Figure 1 molecules-27-05231-f001:**
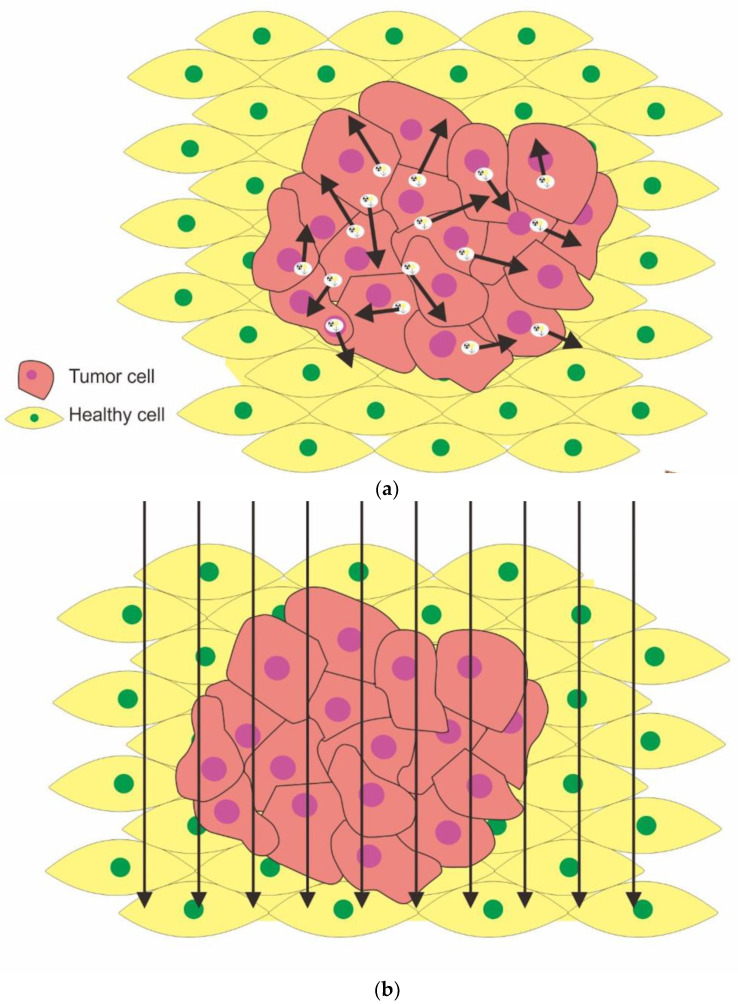
The cell’s radiation distribution by RPT (**a**) and external radiotherapy (**b**). Radiopharmaceuticals are administrated intravenously to be delivered to a target tumor. The targeted tumor cell absorbs a dose of radiation which exponentially decreases over time. The tumor mass’s periphery cells will receive absorbed and crossfire doses from other target cells (**a**). Radiation beams are directed at tumor tissue in external radiotherapy and can also affect healthy cells (**b**).

**Table 1 molecules-27-05231-t001:** Characteristics of radionuclides used in radiotherapy.

Radionuclides	Emitting	Physical Half-Life	Mean Eα/β- (MeV)			Primary Eα/β- (MeV) (%)	Mean Range in Soft Tissue (mm)	Indication	References
			Max	Min	Mean				
** ^131^ ** **I**	β	8.02 d	0.606 MeV	0.069 MeV	0.356 MeV	0.3645 MeV (81%)	0.4 mm	Hyperthyroid, thyroid cancer, Radioimmunotherapy (RIT) for NHL and neuroblastoma, pheochromocytoma, carcinoid, medullary thyroid cancer	[2,3,6,8,22,24]
** ^32^ ** **P**	β	14.26 d	1.71 MeV	0.695 MeV	1.015 MeV	-	2.6 mm	Polycythemia vera, keloid, cystic craniopharyngioma,	[2,3,23]
** ^89^ ** **Sr**	β	50.53 d	1.491 MeV	0.583 MeV	0.908 MeV	0.91 MeV (0.01%)	2.4 mm	Bone pain palliation	[2,3,6,8,23]
** ^90^ ** **Y**	β	64.10 d	2.284 MeV	0.935 MeV	1.349 MeV	(0.01%)	3.6 mm	Liver metastasis, hepatocellular carcinoma, RIT for NHL, neuroendocrine tumor	[2,3,6,8,22,23]
** ^153^ ** **Sm**	β	46.50 h	0.8082 MeV	-	-	0.1032 MeV (29.8%)	0.7 mm	Bone pain palliation, synovitis	[2,3,6,8]
** ^169^ ** **Er**	β	9.4 d	0.35 MeV	-	-	0.084 MeV (0.16%)	0.3 mm	Synovitis	[2,3]
** ^177^ ** **Lu**	β	6.73 d	0.497 MeV	0.047 MeV	0.208 MeV	0.208 MeV (11%)	0.28 mm	Synovitis and RIT for various cancer	[2,6,8,22,23,24]
** ^186^ ** **Re**	β	3.72 d	1.077 MeV	0.308 MeV	0. 769 MeV	0.137 MeV (9.4%)	1.2 mm	Bone pain palliation, arthritis	[2,6,8,23]
** ^188^ ** **Re**	β	17 h	2.12 MeV	0.528 MeV	1.592 MeV	0.155 MeV (15%)	2.1 mm	Bone pain palliation, RIT for various cancer, rheumatoid arthritis	[2,3,8,22,23]
** ^223^ ** **Ra**	α	11.44 d	5.9792 MeV	-	6.59 MeV	0.154 MeV (5.59%)	0.054 mm	Bone pain palliation	[2,5,13]
** ^211^ ** **At**	α	7.2 h	-	-	6.79 MeV	(5.87%)	0.057 mm	RIT leukemia, brain tumor, RLT prostate cancer	[2,3,23,25]
** ^213^ ** **Bi**	α	46 mins	-	-	8.32MeV	(26%)	0.078 mm	RIT leukemia, brain tumor	[3,22,23,25]
** ^225^ ** **Ac**	α	10 d	-	-	0.218MeV	(11.4%)	0.05–0.08 mm	Radioligand (RLT) prostate cancer	[2,8,24]

## Data Availability

Not applicable.

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
