# Peer review of "Radiopharmaceutical Treatments for Cancer Therapy, Radionuclides Characteristics, Applications, and Challenges"

_molecules, 2022, doi:10.3390/molecules27165231_

Round 1

Reviewer 1 Report

The manuscript of Elliyanti and coworkers provides a very good overview of the opportunities and challenges of radionuclide therapy. It presents all important radionuclides and their properties with their advantages and disadvantages and cites the most important studies that already describe successful treatments in humans. The manuscript is well written, understandable, and suitable for publication in “Molecules.” However, there are still a few things to be modified:

Please correct in line 83: “Auger electrons”

Please correct in line 142: Superscript of “4He”

Please correct in line 184-185:”225Ac” (not “255Ac”)

Please correct in line 203: “(77Br)”

Please correct in line 237: This is sentence misleading: “Radionuclides are labelled with monoclonal antibodies …” Not the radionuclide is labelled, but the antibodies are labelled with radionuclides etc.

Please correct in line 325:  “..with ?-emitters”

Please correct in line 327:  “to b-emitters”; “?-particles”

Please correct in 398-399:  1,4,7,10-tetraazacyclodocecane-N,N',N'',N'''-tetraacetic acid”

Please correct in line 419:  “with ?-particles”

Please do not forget to superscript the mass numbers of the nuclides in section 3.2, 3.3, 3.4 and 3.5

Author Response

Dear Reviewer 1,

Thank you very much for your effort and suggestion. We highly appreciate it. We made revisions based on your comments. Please see attachment.

Reviewer 2 Report

The review is structured and written concisely with a good state-of-the-art overview. Overall, it suits the scope of the selected journal and is highly timely to the community.

Changes within the conclusion section may improve the impact of the review to the previously published book chapter (DOI: 10.5772/intechopen.99334). The conclusion part would benefit from more volume regarding future directions or perspectives. Furthermore, I think well-chosen images would not only help the reader to follow, but to lighten the text. This could be for example an overview of discussed structures or the principle of the LET of alpha, beta and Auger.

Comments and remarks:

(1)    The atomic mass of the radionuclide must be superscripted (e.g. line 268 and following) as well as the charges of ions (e.g. line 349 and following,…Na+)

(2)    Line 203: Abbreviation of bromine-77 must be 77Br not 77BRr

(3)    The resolution of the tables is bad. It seems to me that the tables are copied from a different source. If the authors re-used them from a different source they have to cite the source.

(4)    Auger electrons are sometimes written as Äuger electrons. Please check for correct spelling.

(5)    What is N0, N00 in 1,4,7,10-tetraazadodecane- N, N0, N00, N000-tetraacetic acid (DOTA), line 399?

Author Response

Dear Reviewer 2,

Thank you very much for your effort and suggestion. We highly appreciate it. We made revisions based on your comments. We adding future direction and figures principle LET, Alpha, Beta and Auger cause DNA damaged.

The tables we sent with Word files, but the attached tables not good resolution when it attached to the manuscript file.

Please see attached for complete responses

Best regards,

Thank you

Reviewer 3 Report

The review presents fundamental radionuclide properties related to tumor biology and the receptor characteristics of radiopharmaceutical therapy development. The work is well designed, and the article is logically organized and easy to follow. Nevertheless there are some aspects that should be attended, particularly in the nomenclature of the radionuclides. In some parts the atomic number is in superscipt form and in others not. The following guideline is recommended to improve the nomenclature:

Gillings, N., Hjelstuen, O., Ballinger, J., Behe, M., Decristoforo, C., Elsinga, P., … Todde, S. (2021). Guideline on current good radiopharmacy practice (cGRPP) for the small-scale preparation of radiopharmaceuticals. EJNMMI Radiopharmacy and Chemistry6(1). https://doi.org/10.1186/s41181-021-00123-2

Check over page 8 line 281, the sentence starts talking about iodine 131 and after that there is no reference to the molecules radiolabelled with that radionuclide, the citation [60] has no correlation with the text as well.

Author Response

Dear Reviewer 3,

Thank you very much for your effort and suggestion. We highly appreciate it. We made revisions based on your comments below

#

Reviewer comments

Responses

Reviewer 3

The work is well designed, and the article is logically organized and easy to follow. Nevertheless, there are some aspects that should be attended, particularly in the nomenclature of the radionuclides. In some parts, the atomic number is in superscript form and in others not.

We agree. we have changed the nomenclature of some radionuclides already, according to the above reviewers’ comments.

Check over page 8 line 281, the sentence starts talking about iodine 131 and after that, there is no reference to the molecules radiolabelled with that radionuclide, the citation [60] has no correlation with text as well.

We agree. we have accordingly checked the sentence and the citation [60] has been replaced with one more relevant to the topic.